# A Block Minifloat Representation for Training Deep Neural Networks

**Sean Fox, Seyedramin Rasoulinezhad, Julian Faraone, David Boland & Philip Leong**
School of Electrical and Information Engineering
The University of Sydney
Sydney, NSW 2006, AUS
`{first}.{last}@sydney.edu.au`

## Abstract

Training Deep Neural Networks (DNN) with high efficiency can be difficult to achieve with native floating-point representations and commercially available hardware. Specialized arithmetic with custom acceleration offers perhaps the most promising alternative. Ongoing research is trending towards narrow floating-point representations, called minifloats, that pack more operations for a given silicon area and consume less power. In this paper, we introduce Block Minifloat (BM), a new spectrum of minifloat formats capable of training DNNs end-to-end with only 4-8 bit weight, activation and gradient tensors. While standard floating-point representations have two degrees of freedom, via the exponent and mantissa, BM exposes the exponent bias as an additional field for optimization. Crucially, this enables training with fewer exponent bits, yielding dense integer-like hardware for fused multiply-add (FMA) operations. For ResNet trained on ImageNet, 6-bit BM achieves almost no degradation in floating-point accuracy with FMA units that are $4.1 \times (23.9\times)$ smaller and consume $2.3 \times (16.1\times)$ less energy than FP8 (FP32). Furthermore, our 8-bit BM format matches floating-point accuracy while delivering a higher computational density and faster expected training times.

## 1 Introduction

The energy consumption and execution time associated with training Deep Neural Networks (DNNs) is directly related to the precision of the underlying numerical representation. Most commercial accelerators, such as NVIDIA Graphics Processing Units (GPUs), employ conventional floating-point representations due to their standard of use and wide dynamic range. However, double-precision (FP64) and single-precision (FP32) formats have relatively high memory bandwidth requirements and incur significant hardware overhead for general matrix multiplication (GEMM). To reduce these costs and deliver training at increased speed and scale, representations have moved to 16-bit formats, with NVIDIA and Google providing FP16 (IEEE-754, 2019) and Bfloat16 (Kalamkar et al., 2019) respectively. With computational requirements for DNNs likely to increase, further performance gains are necessary in both datacenter and edge devices, where there are stricter physical constraints.

New number representations must be easy to use and lead to high accuracy results. Recent 8-bit floating-point representations have shown particular promise, achieving equivalent FP32 accuracy over different tasks and datasets (Wang et al., 2018; Sun et al., 2019). We refer to such representations as *minifloats* in this paper. Minifloats are ideal candidates for optimization. By varying the number of exponent and mantissa bits, many formats can be explored for different trade-offs of dynamic range and precision. These include logarithmic and fixed point representations which provide substantial gains in speed and hardware density compared to their floating-point counterparts. For instance, 32-bit integer adders are approximately $10\times$ smaller and $4\times$ more energy efficient than comparative FP16 units (Dally, 2015). That said, fixed point representations still lack the dynamic range necessary to represent small gradients for backpropagation, and must be combined with other techniques for training convergence.

Block floating point (BFP) in (Yang et al., 2019; Drumond et al., 2018) share exponents across blocks of 8-bit integer numbers, and provide a type of coarse-grained dynamic range for training. This

approach will typically incur some accuracy loss on more challenging datasets, however all dot-products within the block can be computed with dense fixed point logic. In comparison, HFP8 (Sun et al., 2019) minifloats require larger floating-point units (expensive FP16 adders in particular) but have at least 5 exponent bits dedicated to each gradient and suffer zero degradation in training accuracy. It would seem that an ideal representation should bridge the gap between each of these approaches. Our work achieves this for 8-bit and sub 8-bit precision schemes, overcoming two key challenges in the process. These are listed below and discussed with related works.

## 1.1 CHALLENGES AND RELATED WORK

**Minimising data loss with fewer bits:** While several works have demonstrated training with fewer than 8 bits of precision, they typically lead to loss of accuracy on more complex problems and have performance bottlenecks because parts of the algorithm are left in high precision (Hubara et al., 2017; Zhou et al., 2016; Miyashita et al., 2016). Therefore, training end-to-end with reduced precision representations that are persistent remains a key challenge. In this regard, 8 bit tensors with 16-bit updates can be trained effectively (Banner et al., 2018). Data loss arises when formats do not have enough range to capture variations in tensor distributions during training. BFloat (Kalamkar et al., 2019) adds two extra exponent bits for a custom 16-bit representation, and the Apex library is used in (Micikevicius et al., 2017; Wang et al., 2018; Sun et al., 2019) for scaling the loss function into a numerically representable range. Block floating point and other variants apply similar functionality for fixed point numbers, but at a finer granularity. WAGE (Wu et al., 2018) uses layer-wise scaling factors, SWALP (Yang et al., 2019) shares exponents across feature maps and convolution channels, and HBFP (Drumond et al., 2018) does the same for dot-products, though their implementation requires caching of intermediate activations in FP32 and wide weight storage for better accuracy. S2FP8 (Cambier et al., 2020) replaces loss-scaling in FP8 (Wang et al., 2018) with squeeze and shift factors that center 8-bit minifloats over the mean exponent of the value distribution. Shift factors operate similarly to BFP shared exponents, whereas squeeze factors can divert precision away from high value regions leading to errors in dot-product calculations. We provide some empirical evidence of this effect in Section 4.5. Finally, HFP8 (Sun et al., 2019) defines two minifloat formats that are optimized for range and precision requirements of forward and backward paths separately. In this work, we seek minifloat formats that are also optimized for arithmetic density.

**Increasing the performance density of floating-point:** Most DNN training frameworks are developed with GEMM accumulation in FP32. The authors in (Wang et al., 2018) reduced the accumulation width to FP16 with chunk-based computations and stochastic rounding. However, training minifloats with even denser dot-products has not been demonstrated. For DNN inference, ELMA (Johnson, 2018) and posit number systems (Gustafson & Yonemoto, 2017) describe arithmetic that accumulate minifloat-like numbers as integers. Such work is applicable when the number of exponent bits is small, however training under such regimes can lead to data loss due to limited dynamic range.

## 1.2 CONTRIBUTIONS

In this paper, we present the Block Minifloat (BM) representation which addresses both of the aforementioned challenges. BM is a modification of block floating point that replaces the fixed point values with minifloats, whilst maintaining shared exponents across blocks of numbers. BM formats generalise a far wider spectrum of reduced precision representations and produce better outcomes than previous 8-bit regimes. Specific contributions of our work include:

- Block Minifloat (BM), a more efficient alternative to INT8 and FP8 for end-to-end DNN training. Shared exponent biases provide dynamic range and accuracy, while small exponent encodings provide fine-grained dynamic range and reduce the hardware cost of GEMM accumulation.

- A new 8-bit floating-point format that uses no more than **4 exponent bits**, achieving equivalent accuracy to floating-point with denser hardware via efficient Kulisch accumulation.

- An exploration of the BM design space showing high accuracy DNN training with sub 8-bit representations for all weights, activations and gradients. This includes two techniques for minimising data loss of a practical implementation, namely gradual underflow and cost-aware block designs.

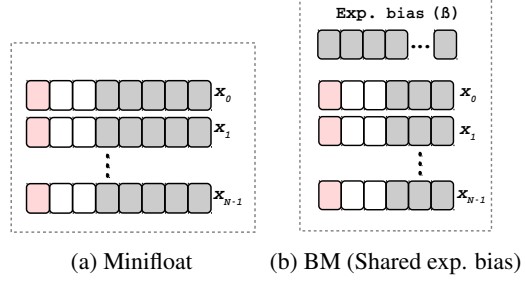

(a) Minifloat  (b) BM (Shared exp. bias)

Figure 1: Minifloat and Block Minifloat (BM) tensor representations

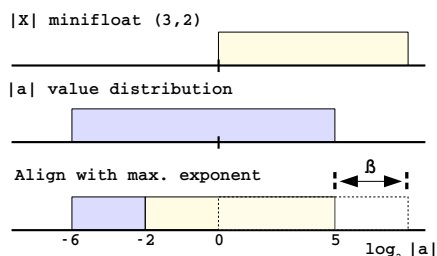

Figure 2: Exponent bias shifts the minifloat distribution to align with the maximum exponent of the value distribution

## 2 BLOCK MINIFLOAT REPRESENTATION

### 2.1 MINIFLOAT NUMBER FORMAT

Equation (1) computes the real value of a minifloat number, where $(e, m)$ denote the number of exponent and mantissa bits in the representation.

$$X\langle e, m\rangle = \begin{cases} E = 0, & (-1)^s \times 2^{1-\beta} \quad \times (0 + F \times 2^{-m}) & \text{(denormal)} \\ \text{otherwise}, & (-1)^s \times 2^{E-\beta} \quad \times (1 + F \times 2^{-m}) & \text{(normal)} \end{cases} \tag{1}$$

The decimal expansions of the exponent and mantissa are both unsigned integers, given by $E$ and $F$ respectively, $s$ refers to the sign bit and $\beta = 2^{e-1} - 1$ is the exponent bias for the binary-offset encoding scheme. This is consistent with IEEE-754 floating-point standards, except that our minifloats are considerably smaller (4-8 bits only), can generalise to multiple $(e, m)$ configurations, and do not handle nan/infinity bit patterns. Instead, arithmetic is allowed to saturate at the limits of the representable range $[X_{min}^+, X_{max}^+]$. For example, a minifloat representation with $X\langle 4, 3\rangle$ have exponent and mantissas that range between $[0, 15]$ and $[0, 7]$ respectively. Therefore, the largest normal and smallest denormal positive numbers are $X_{max}^+ = 480$ and $X_{min}^+ = 2^{-9}$. This corresponds to a dynamic range of 108 dB shown in Table 7 in Appendix A.1.

### 2.2 SHARED EXPONENT BIAS

The main difference between minifloat and block minifloat (BM) representations are highlighted in Figure 1. Minifloats have one exponent per element, but that exponent must be wide enough to tolerate changes in DNN tensor distributions during training (i.e. 5 bits for gradients in FP8 (Wang et al., 2018)). In contrast, BM share exponent biases across blocks of $N$ minifloat numbers. This provides equivalent dynamic range with respect to the block, but with fewer exponent bits than the original minifloat format. Block floating point (BFP) operates similarly, but all numbers within the block are integers (Drumond et al., 2018). BM can generalise for this case, i.e. when $e = 0$.

The real value of the $i^{th}$ element from BM tensor $a$ is given in Equation (2), where $X$ is an unbiased minifloat tensor, represented by $(e,m)$ exponent and mantissa bits, and $\beta_a$ is the shared exponent bias.

$$a_i = X_i\langle e, m\rangle \times 2^{-\beta_a} \tag{2}$$

In this example, $a_i$ can only be represented accurately when the shared exponent bias $\beta_a$ (calculated for the entire tensor) and the distribution of $X$ jointly captures the value distribution of $a$. For example, large and small values in $a$ could saturate or be lost altogether if $\beta_a$ is too large or too small. However, some leeway exists when exponents are shared across dot products. This is because dot products are reduce operations, meaning their sum is dominated by the largest values in the inputs. For this reason, we calculate $\beta_a$ to specifically guard against overflow, and unlike (Cambier et al., 2020) we don't apply any scaling which could divert precision away from larger value regions. Our method of updating $\beta$ during training is illustrated in Figure 2 and formalized in Equation 3 below.

$$\beta_a = \max\left(\lfloor\log_2 |a|\rfloor\right) - \left(2^e - 1\right) \tag{3}$$

The first term denotes the maximum exponent for the tensor $a$, which changes and must be updated during training, while the second term is fixed and refers to the maximum exponent of $X$.

In terms of hardware, shared biases ensure that all dot products can be computed with denser minifloat arithmetic. This is shown in Equation 4 for BM tensors $a$ and $b$, each with $N$ elements.

$$a \cdot b = \sum_{i=1}^{N} \left( (X_i^a \times 2^{-\beta_a}) \times (X_i^b \times 2^{-\beta_b}) \right) = 2^{-(\beta_a + \beta_b)} \times (X^a \cdot X^b) \tag{4}$$

The dot product, $X^a \cdot X^b$, have minifloat formats with smaller exponents, while the cost of calculating, storing and aligning the exponent biases during training is amortized over the length of the dot-product. Next we show how minifloat formats with fewer exponent bits lead to faster and more compact hardware.

## 2.3 KULISCH ACCUMULATION

A Kulisch accumulator (Kulisch & Miranker, 2014) is a fixed point accumulator that is wide enough to compute an error free sum of scalar floating-point products, over the entire range of possible values. Kulisch accumulators operate by shifting the mantissa of the floating-point product into an internal register according to the exponent of the product. The sum proceeds as integer addition which is $4-10\times$ more efficient in terms of area and power compared to FP16 (Dally, 2015). The number of bits required for the internal register (i.e. the addend) and shifter, scale the size and complexity of the accumulator and are provided as formulas in Equation (5) for BM operands $a = (e_a, m_a)$ and $b = (e_b, m_b)$.

Table 1: Kulisch accumulator examples

| Format | Kulisch Acc. | |
| --- | --- | --- |
| $(e_a, m_a)/(e_b, m_b)$ | kadd | kshift |
| $(8, 23)/(8, 23)$ | 561 | 512 |
| $(5, 2)/(6, 1)$ | 102 | 96 |
| $(4, 3)/(5, 2)$ | 56 | 48 |
| $(3, 4)/(4, 3)$ | 34 | 24 |
| $(2, 3)/(3, 2)$ | 20 | 12 |
| INT8 | 32 | - |

$$kadd = 1 + (2^{e_a} + m_a + 1) + (2^{e_b} + m_b + 1) \tag{5}$$
$$kshift = 2^{e_a} + 2^{e_b} \tag{6}$$

In the above equations, *kadd* calculates the number of bits required for the largest product of two numbers, plus one extra bit for the addition, and *kshift* determines the maximum number of bits the mantissa product must be shifted to align with the addend. Crucially, by considering the size of $kadd$ and $kshift$, BM formats can be designed to trade-off fine-grained dynamic range (i.e. exponent bits) for more precision and smaller hardware. In fact, formats with exponents up to 4 bits may yield *kadd* of approximately the same size as INT8/INT32 arithmetic units, while *kadd* becomes prohibitively wider and more expensive for larger exponents. This is clearly shown via example in Table 1 above, but more importantly, it is supported by hardware synthesis results given in Section 5 and Appendix A.4. For example, an 8-bit minifloat format having 4 exponent bits achieves a $1.6\times$ area reduction compared to HFP8 (Sun et al., 2019) with 5 exponent bits. Furthermore, through an extensive set of experiments we discover that such representations can also achieve high training accuracy which forms a key contribution of our work.

## 3 TRAINING WITH BLOCK MINIFLOAT

### 3.1 MINIMIZING DATA LOSS

BM arithmetic will incur data loss when the value distribution is too wide or requires more precision than can be captured by the underlying minifloat representation within a block. Below, we describe steps taken to mitigate this problem without substantially increasing implementation overheads.

**Gradual underflow:** Our minifloats support denormal numbers as defined in Equation (1). Denormal numbers have precision close to zero, and ensure that consecutively smaller quantized numbers approach zero gradually. The alternative is flush-to-zero which discards the mantissa bits when $E = 0$. This equates to approximately 12.5% of the exponent encoding when $e = 3$; this is highly inefficient. Overhead for denormal numbers in hardware is minimal, and only requires detection of $E = 0$ and a single bit flip in the multiplier. Our experiments show that gradual underflow is crucial for BM formats with less than four exponent bits.

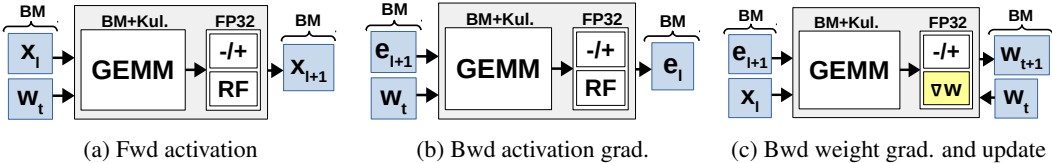

(a) Fwd activation        (b) Bwd activation grad.        (c) Bwd weight grad. and update

Figure 3: End-to-end Training with Block Minifloat (BM). All off-chip memory transfers are low precision BM tensors. BM alignments, weight updates, quantization, batchnormalization and ReLU are executed in on-chip scalar FP32 units. The register file (RF) stores a block of $\nabla W$.

**Block Size:** Matrix multiplication with BM is computed by dividing tensors into $N \times N$ blocks that bound the number of exponent biases and reduce data loss, since each block shares one exponent bias. Square blocks are chosen so that biases are contiguous in memory regardless of whether the block is operating in the forward path or after transposition in the backward path. As such, BM can be stored with a persistent data structure, that doesn't require recasting or extra memory transfers during training. This makes BM easy to use at the software level, but does mean that biases are shared across $N$ independent dot products. In terms of hardware cost, Equation 7 formalizes the relationship between the size of $N$ and three overheads, where $\alpha$ refer to relative area costs for each overhead.

$$\text{cost} = \overbrace{\alpha_1\big(1 + \frac{\log_2 N}{\text{kadd}}\big)}^{\text{Kulisch}} + \overbrace{\alpha_2\frac{1}{N}}^{\text{FP}} + \overbrace{\alpha_3\frac{8 + N^2}{N^2}}^{\text{Memory}} \tag{7}$$

For the first term, the width of the Kulisch accumulator must increase by $\log_2 N$ bits to prevent overflow in the BM dot-products. In the second term, floating-point hardware (including quantization and conversion modules) are required to accumulate, align and convert BM partial results, but the cost is amortized over $N$ fixed point operations. Finally, in the last term, additional memory is required to compute and store one 8-bit bias every $N \times N$ values. For large block sizes, the extra silicon area from Equation 7 is negligible compared to the GEMM but data loss from sharing biases can still be significant. In Section 4.5, we used Equation 7 and determined that a block size of $N = 48$ offers a good balance for both objectives and is used for the rest of this paper.

**Hybrid representation:** Different minifloat representations for forward and backward paths have been shown to produce better accuracy for FP8 training (Sun et al., 2019). We apply the same idea, and find the best balance of precision and range for both paths separately. Full details for all BM formats are provided in Table 7 (in Appendix A.1) where forward and backward configurations are given by $(e_f, m_f)/(e_b, m_b)$ notation. Our formats cover each precision level between 4 and 8 bits, and are denoted by BM4, BM5, BM6, BM7 and BM8 in our experiments. For example, BM6 $(2,3)/(3,2)$ refers to 6-bit BM training with weight and activation tensors represented by $(2,3)$ and activation gradient tensors represented by $(3,2)$ minifloat formats.

### 3.2 TRAINING DETAILS AND GPU SIMULATION

BM offers an alternative to standard FP32 for the computationally intensive parts of training, which is typically mapped to general matrix multiplication (GEMM). However, specialised hardware is required to realise its potential gains in speed and energy efficiency. For the purposes of this paper, we simulate the behaviour of BM hardware using GPUs and PyTorch [1]. Given that dot products are computed exactly via Kulisch accumulators, existing CUDA libraries for GEMM can be used without modification, and all data loss is attributed to quantization only. Figure 3 provides an illustration of the setup for each GEMM in forward and backward paths. In a practical implementation, BM does not require any costly movement or storage of high precision tensors. This is enabled by scalar processors after the GEMM (for FP32 operations, Kulisch to floating-point conversion, block minifloat alignments, quantization etc.) and a weight update scheme that can compute and cache high precision gradients on-chip (Sun et al., 2019). Weight, activation and gradient tensors are quantized to BM numbers with stochastic rounding as described in (Wang et al., 2018). For the software simulation, quantization is applied before each GEMM in forward and backward paths and contributes significant performance overhead compared to standard PyTorch layers. An approximate

---

[1]Our implementation is available at https://github.com/sfox14/block_minifloat

$5\times$ slow-down is realised on most networks and datasets, with support for denormal numbers the main implementation bottleneck. The realisation of the same function is comparatively cheap in custom hardware however, and can be fully-pipelined for fast training times.

## 4 EXPERIMENTS

We evaluated the training accuracy of BM on a subset of image, language and object detection modelling tasks. The entire spectrum of representations were explored on ImageNet (Deng et al., 2009) and CIFAR (Krizhevsky et al., 2009) image recognition benchmarks, with results compared against well-calibrated INT8, FP8 and FP32 baselines. On other tasks, BM8 is compared with an FP32 baseline.

Table 2: Final Validation Accuracy (%) on CIFAR datasets for ResNet-18

| Scheme | CIFAR-10 | CIFAR-100 |
|---|---|---|
| FP32 (ours) | 94.9 | 77.5 |
| BM6 (2,3)/(3,2) | 95.1 | 77.2 |
| BM5 (2,2)/(3,1) | 94.7 | 76.1 |
| BM4 (2,1)/(3,0) | 94.2 | 73.7 |

Table 3: Training Accuracy (%) on CIFAR-10 for VGG16 and log quantization

| CIFAR-10 | FP32 | Log | $\nabla$ | kshift (bits) |
|---|---|---|---|---|
| log-5b[1] | 94.1 | 93.8 | -0.3 | 32 |
| log-BM5 (ours) | 93.8 | 93.4 | -0.4 | 32 |
| log-BM4 (ours) | 93.8 | 93.1 | -0.7 | 16 |

[1] (Miyashita et al., 2016)
[2] results achieved with base $\sqrt{2}$

### 4.1 CIFAR-10 AND CIFAR-100

We ran CIFAR experiments using SGD with momentum of 0.9 for 200 epochs in batches of 128 images and initial learning rate of 0.1 which is decayed by a factor of 5 at the 60th, 120th and 160th epochs. Table 2 presents results for training ResNet-18 (He et al., 2016) with only small BM6, BM5 and BM4 representations. These offer the highest reduction in memory usage while still reaching very close to the FP32 baseline. For example, 6-bit BM training only records a 0.3% loss in accuracy compared to FP32 on CIFAR-100 while theoretically saving 25% of memory read and write overheads compared to FP8. We also tested logarithmic BM formats on CIFAR-10 and VGG16 network. Log representations arise when $m = 0$, and require only adds and shifts for multiply-add arithmetic. Our results use the same training parameters as before and are shown in Table 3. We compare against the only previously known result for log training, i.e. *log-5b* (Miyashita et al., 2016) and achieve similar results with respect to FP32 for 5-bit and 4-bit. BM representations have exponent biases that shifts tensor distributions dynamically during training, whereas *log-5b* define offset parameters at each layer that are fixed. Allowing biases to vary during training gives BM an advantage, and results in similar validation accuracy with only 4-bit words. This corresponds to approximately half the cost for multiplication in the linear domain (by exponent add and Kulisch shift).

Table 4: Top-1 accuracy (%) of reduced precision (RP) training on ImageNet for ResNet-18 models

| Scheme | Numerical representation $(e, m)$ | | | | | ResNet-18 | |
|---|---|---|---|---|---|---|---|
| | w | x | dw | dx | acc | FP32 | RP |
| SWALP (Yang et al., 2019) | $8^{1}$ | $8^{1}$ | $8^{1}$ | $8^{1}$ | $32^{1}$ | 70.3 | 65.8 |
| S2FP8 (Cambier et al., 2020) | $(5, 2)/(8, 23)$ | $(5, 2)$ | $(5, 2)$ | $(5, 2)$ | $(8, 23)$ | 70.3 | 69.6 |
| HFP8 (Sun et al., 2019) | $(4, 3)$ | $(4, 3)$ | $(5, 10)$ | $(5, 2)$ | $(5, 10)$ | 69.4 | 69.4 |
| BM8 (2,5)/(4,3) | $(2, 5)$ | $(2, 5)$ | $(6, 9)$ | $(4, 3)$ | $31^{1}$ | 69.7 | 69.8 |
| BM7 (2,4)/(4,2) | $(2, 4)$ | $(2, 4)$ | $(6, 9)$ | $(4, 2)$ | $29^{1}$ | 69.7 | 69.6 |
| BM6 (2,3)/(3,2) | $(2, 3)$ | $(2, 3)$ | $(6, 9)$ | $(3, 2)$ | $20^{1}$ | 69.7 | 69.0 |
| BM5 (2,2)/(3,1) | $(2, 2)$ | $(2, 2)$ | $(6, 9)$ | $(3, 1)$ | $18^{1}$ | 69.7 | 66.8 |

[1] Fixed point

## 4.2 IMAGENET

The ImageNet dataset has 1000 class labels, and consists of 256x256 images split into a training set with 1.28 million images and validation set with 50,000 images. We use ResNet-18 (He et al., 2016) and AlexNet (Krizhevsky et al., 2012) architectures from the official PyTorch implementation [2], and train on one GPU with standard settings; SGD with momentum of 0.9, batches of 256 images, and an initial learning rate of 0.1 (0.01 for AlexNet) which is decayed by a factor of 10 at epoch 30 and 60. ResNet-18 has been widely tested upon in previous work, and offers the most suitable benchmark for exploring the full spectrum of BM representations, especially given the size of the network as well as the cost of BM quantization on training times (approx. 5× slow-down). Results are presented in Table 4 where columns w, x, dw, dx and acc refer to the numerical representation for weight, activation, weight gradient, activation gradient and on-chip GEMM accumulator. We achieve FP32 equivalent accuracy for BM8 and BM7, slight degradation for BM6, while our BM5 exceeds the reported accuracy for 8-bit SWALP (Yang et al., 2019). Compared to S2FP8 (Cambier et al., 2020), our BM6 representation reaches similar levels of relative accuracy, but with two fewer bits and without a high precision master copy of the weights. We provide some insights into possible reasons for this in Section 4.5 by considering the possibility of diminishing returns in accuracy from scaling minifloat representations. Compared with HFP8 (Sun et al., 2019), which offers robust 8-bit training results, BM8 produces the same accuracy on ImageNet while improving upon HFP8 in hardware density and performance. BM8 tensors can be represented with fewer exponent bits, and thus perform dot products via Kulisch accumulators that are smaller and faster than FP16 units. Furthermore, BM offers tradeoffs for even denser arithmetic and lower memory usage. In these regimes, BM hardware is more comparable to SWALP (Yang et al., 2019) which performs the GEMM in fixed point. Proof of BM design efficiencies are provided with RTL synthesis results in Section 5, Figure 6.

| Model (Dataset) [Metric] | FP32 | BM8 |
|---|---|---|
| AlexNet (ImageNet) | 56.0 | 56.2 |
| EfficientNet-b0 (small ImageNet) | 62.6 | 61.8 |
| LSTM (PTB)[Val ppl.] | 84.7 | 87.33 |
| Transformer-base (IWSLT)[BLEU] | 32.3 | 31.8 |
| SSD-Lite (MbNetV2) (VOC)[mAP] | 68.6 | 68.0 |

Table 5: Baseline FP32 v BM8 training on Image, Language and Object Detection models

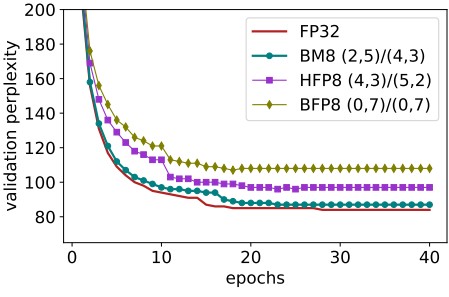

Figure 4: Validation perplexity of LSTM model on Penn Treebank

## 4.3 LANGUAGE MODELLING WITH LSTM

We compared 8-bit formats for language modeling on the Penn Treebank dataset Marcus et al. (1993). We adapted the 2-layer Long Short Term Memory (LSTM) network from PyTorch Examples [3] and perform all GEMM operations with BM8 arithmetic. The batch size is 20, initial learning rate is 20 with 0.25 decay, the embedding and hidden dimensions are 650 and sequence length is 35. Results in Figure 4 show BM8 with $(2, 5)/(4, 3)$ hybrid configuration achieving better accuracy than BFP8 and HFP8 variants. The proposed BM8 representation has more fine-grained dynamic range and fewer mantissa bits than BFP8, and more precision and fewer exponent bits than HFP8 formats. This design point achieves better outcomes in terms of accuracy and hardware density than either representation separately (see Figure 6, Section 5). Validation perplexity of 87.33 is also comparable to 84.70 obtained with full-precision floating-point.

## 4.4 ADDITIONAL EXPERIMENTS

To demonstrate wider applicability of the BM number representation, we tested BM8 on several additional networks and modelling tasks. Results are summarized in Table 5 with full details of each

---

[2]Implementation available at https://github.com/pytorch/examples/tree/master/imagenet

[3]Implementation available at https://github.com/pytorch/examples/tree/master/word_language_model

experiment provided in Appendix A.3. Crucially, every network tested achieved comparable accuracy with baseline FP32. This includes EfficientNet-b0 (Tan & Le, 2019) image classification and SSD-lite (Liu et al., 2016) with MobileNet-V2 object detection models, both of which represent the type of network and application well suited to resource constrained hardware devices. Furthermore, we also trained a small Transformer network for translation on the IWSLT German to English dataset (Cettolo et al., 2014). In future work, we plan to scale our implementation and demonstrate training with BM representations on larger networks and datasets. Network design with BM is another interesting research direction, since the majority of network architectures have been designed and optimized while assuming an FP32 arithmetic scheme.

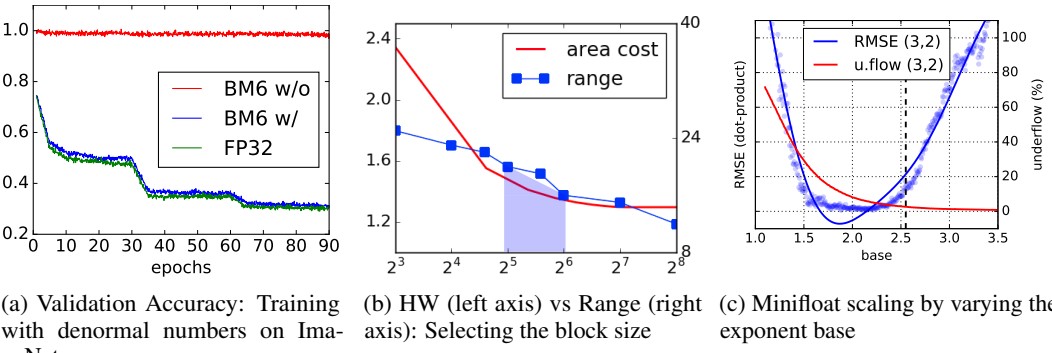

(a) Validation Accuracy: Training with denormal numbers on ImageNet

(b) HW (left axis) vs Range (right axis): Selecting the block size

(c) Minifloat scaling by varying the exponent base

Figure 5: Experiments for minimising data loss with 6-bit Block Minifloat (BM6)

## 4.5 EMPIRICAL ANALYSIS

**Effect of Denormal Numbers:** To study the effect that denormal numbers have on training convergence in sub 8-bit networks, we trained ResNet-18 on ImageNet for BM6 with denormals (ours) and without denormals, using QPyTorch library (Zhang et al., 2019). Results are plotted against floating-point accuracy in Figure 5a. Without denormals, small numbers are flushed-to-zero and training stagnates immediately. Although not shown here, 8-bit representations with more than $e = 3$ bits do not suffer similar accuracy degradation without denormals. This investigation confirms the importance of denormal numbers for training BM formats with fewer exponent bits, and differentiates our software and hardware experiments substantially from previous 8-bit regimes.

**Selecting the Block Size:** We conducted experiments on CIFAR100 to determine suitable block sizes - those which simultaneously increase dynamic range and have low hardware overhead. Results are shown in Figure 5b. We took the average of the largest range observed in gradient tensors at different block settings, over the entire duration of training. Estimates of area come from Equation 7 with parameters; $\alpha_1 = 1$, $\alpha_2 = 10$ (relative area of fixed point and floating-point respectively), $kadd = 21$ and $\alpha_3 = 0$. We saturate the area cost at $N = 256$, which is consistent with the length of dot-products supported by the GEMM architecture in TPU hardware (Jouppi et al., 2017). Finally, $N = 48$ emerged as a good selection, corresponding to one floating-point unit every 48 multiply-accumulate operations and one 8-bit exponent bias every 2304 minifloat numbers.

**Scaling the Minifloat Representation:** In Figure 2, which was discussed previously, minifloats have exponent biases that shift the representation to align with the maximum of the underlying value distribution. Additionally, the minifloat representation could be scaled (or stretched) over a wider or narrower part of the value distribution. We investigate this effect by varying the base of the exponent, and inspecting the underflow and root mean square error (rmse) of dot-products after quantization; results are shown in Figure 5c. The tensor under test is a gradient tensor with maximum exponent of -17 and mean exponent of -21. Mean scaling was proposed in S2FP8 (Cambier et al., 2020) for 8-bit training and works by centering the minifloat over the mean of the exponent value distribution. For the (3,2) format, mean scaling requires a base of 2.52, calculated as $b = 2^{\frac{-17+21}{7-4}}$. This is akin to redirecting precision from high value regions into smaller underflow regions, the result of which observably leads to increased error in the tested 6-bit regime. Better approaches could be designed to detect underflow and use higher precision arithmetic where necessary.

| Component | Area $(\mu m^2)$ | Power $(\mu W)$ |
|---|---|---|
| FP32 | 4782 | 10051 |
| FP8 (w/ FP16 add) | 829 | 1429 |
| INT8 (w/ INT32 add) | 417 | 1269 |
| BM8 | 391 | 1141 |
| BM6 | **200** | **624** |
| INT8 (4x4 systolic) | 7005 | 20253 |
| FP8 (4x4 systolic) | 18201 | 56202 |
| BM8 (4x4 systolic) | **6976** | **18765** |

Table 6: Logic area and power of single-cycle fused multiply-Add (FMA) and 4x4 array multipliers. Synthesized at 750 MHz with Cadence RTL Compiler 14.11 and 28nm cell library

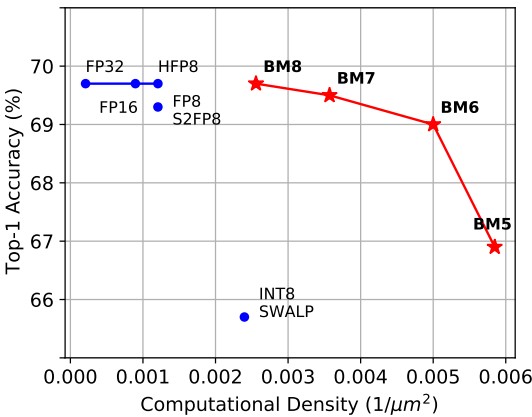

Figure 6: Computational density v ResNet-18 accuracy on ImageNet

## 5 HARDWARE EVALUATION

In this section, we evaluate the proposed block minifloat representation in hardware and compare against competitive integer and floating-point arithmetic. Figure 6 summarizes our results with a plot of computational density (measured as operations per unit silicon area) and ResNet-18 training accuracy on ImageNet. Computational density was obtained from an RTL design of single-cycle fused multiply-add (FMA) units and 4x4 systolic array multipliers. We performed synthesis at 750MHz for 28nm silicon technology and recorded area and power measurements for each number representation. Table 6 provides a subset of these results, with coverage of all BM formats supplied in Appendix A.4.

In summary, BM8 and BM6 arithmetic units are $2.1 \times (12.2\times)$ and $4.1 \times (23.9\times)$ smaller and consume $1.25 \times (8.8\times)$ and $2.3 \times (16.1\times)$ less power than competitive FP8/(FP32) representations. Such arithmetic, which has similar hardware complexity to INT8, may be especially useful in embedded applications where there are stricter area and power constraints but training still needs to achieve normal levels of accuracy and relatively high performance. With high computational density, BM arithmetic can achieve higher training throughput on compute intensive problems, while sub 8-bit BM formats have lower bandwidth requirements leading to faster training times in memory bound applications. Finally, overheads related to conversion from Kulisch to floating-point and BM quantization are expected to contribute little logic area relative to GEMM. This includes modules for leading-one detection, barrel shifter, maximum exponent calculation, pre-quantization buffering and stochastic rounding, each of which have an efficient implementation. Further support of these claims and other system-level effects are the subject of future work.

## 6 CONCLUSION

A new representation called Block Minifloat (BM) was presented for training DNNs effectively with reduced precision. Our representation allows the implicit exponent bias within IEEE-754 floating-point specifications to vary for a block of numbers, and can be trained with high accuracy using narrow exponent encodings. We describe how few exponent bits lead to significantly smaller hardware, while smaller representations reduce memory bandwidth requirements, leading to faster training than previous 8-bit approaches.

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

# A  Appendix

## A.1  Block Minifloat Number Formats

The full spectrum of Block Minifloat (BM) formats are presented in Table 7. BM formats are designed with consideration for the hardware cost of the Kulisch accumulator and also dynamic range and precision requirements.

Table 7: Comparison of Block Minifloat number formats

| Scheme | Format[1] $(e, m)$ | Range[2] (dB) | Precision[3] $(\epsilon)$ | Kulisch Acc. kadd | kshift |
|---|---|---|---|---|---|
| FP32 | (8,23) | 1668 | $2^{-24}$ | 561 | 512 |
| Bfloat16 (Kalamkar et al., 2019) | (8,7) | 1529 | $2^{-8}$ | 529 | 512 |
| FP16 | (5,10) | 241 | $2^{-11}$ | 87 | 64 |
| FP8 (Wang et al., 2018) | (5,2) | 185 | $2^{-3}$ | 71 | 64 |
| S2FP8 (Cambier et al., 2020) | (5,2) | 185 | $2^{-3}$ | 71 | 64 |
| HFP8 (Sun et al., 2019) | (4,3)/(5,2) | 108/185 | $2^{-4}/2^{-3}$ | 56 | 48 |
| SWALP (Yang et al., 2019) | (0,7) | 42.1 | $2^{-8}$ | 32 | - |
| INT8 (Wu et al., 2018) | (0,7) | 42.1 | $2^{-8}$ | 32 | - |
| BM8 (ours) | (2,5)/(4,3) | 48.0/108 | $2^{-6}/2^{-4}$ | 31 | 20 |
| BM7 (ours) | (2,4)/(4,2) | 41.9/101 | $2^{-5}/2^{-3}$ | 29 | 20 |
| BM6 (ours) | (2,3)/(3,2) | 35.6/53.0 | $2^{-4}/2^{-3}$ | 20 | 12 |
| BM5 (ours) | (2,2)/(3,1) | 28.9/45.7 | $2^{-3}/2^{-2}$ | 18 | 12 |
| BM5-log (ours) | (4,0)/(4,0) | 28.9/45.7 | $2^{-3}/2^{-2}$ | 33 | 32 |
| BM4 (ours) | (2,1)/(3,0) | 21.6/42.1 | $2^{-3}/2^{-1}$ | 16 | 12 |
| BM4-log (ours) | (3,0)/(3,0) | 28.9/45.7 | $2^{-3}/2^{-2}$ | 17 | 16 |

[1] hybrid formats, i.e. forward/backward
[2] dynamic range in decibels $20 \log_{10}(X_{max}^+ / X_{min}^+)$
[3] relative round-off error, i.e. $2^{-m} \times 2^{-1}$

## A.2  Software Implementation Details

Block Minifloat (BM) arithmetic requires custom hardware to achieve gains in speed and energy efficiency. We use QPyTorch (Zhang et al., 2019), an open source framework for low-precision training, to simulate the behaviour of BM hardware with existing PyTorch and CUDA libraries. QPyTorch provides a simple interface for applying quantization in the forward path (weight and activation tensors) and backward path (error, gradient and momentum tensors), ensuring that all numbers have a low-precision representation, while the actual GEMM and AXPY operations are computed in single-precision floating-point (FP32). This last point means that QPyTorch can not ordinarily be used to research low-precision accumulation strategies. Our work is different. Kulisch accumulators (as described in Section 2.3) compute exact dot-products, and therefore our GEMMs are adequately approximated by FP32 arithmetic (which is close to exact). QPyTorch is avalaible online [4] and supports floating-point (without denormals), fixed point and block floating point number formats. Our code implementation[5] is an extension of this package.

---

[4] Implementation available at https://github.com/Tiiiger/QPyTorch
[5] Available at https://github.com/sfox14/block_minifloat

### A.3    MODEL DETAILS AND ADDITIONAL EXPERIMENTS

#### A.3.1    BLOCK MINIFLOAT CONVERGENCE CURVES ON IMAGENET

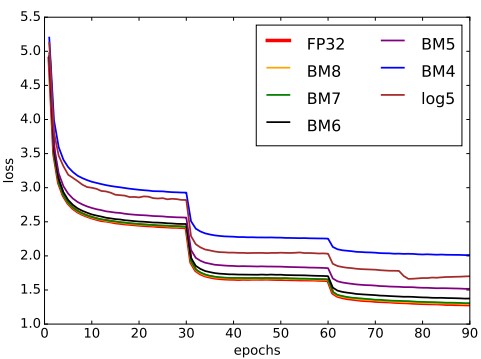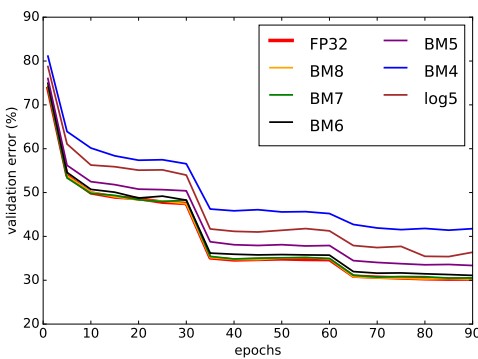

Figure 7: Train loss and top-1 validation accuracy for the full spectrum of Block Minifloat formats trained on ImageNet using a ResNet-18 model

#### A.3.2    COMPARISON WITH BLOCK FLOATING POINT (BFP) ON IMAGENET

As described in Section 1, block minifloats bridge the gap between narrow floating-point and block floating point (BFP) representations. The main idea is that better outcomes in terms of accuracy and hardware efficiency can be achieved by exploring the spectrum between the two representations. While BFP ensures that the majority of computation involves dense integer arithmetic, the lack of fine-grained dynamic range typically leads to accuracy loss on larger models and more complex datasets. In Table 8 (below), we show that BM recovers accuracy loss for 6-bit and 8-bit formats on ImageNet while maintaining the same advantages in hardware.

Table 8: Comparison of Block Minifloat (BM) and Block Floating Point (BFP) number formats trained on ImageNet with ResNet-18 model.

| Scheme | BFP (ours) | BM (ours) | $\nabla$ |
|--------|-----------|-----------|----------|
| 6-bit | 67.0 | 69.0 | +2.0 |
| 8-bit | 69.2 | 69.8 | +0.6 |

#### A.3.3    TRANSFORMER MODEL (WMT)

We trained the Transformer Base model from the FairSeq [6] repository on the IWSLT'14 German to English translation task. We used Adam optimizer and modified the FairSeq implementation with BM8 quantization. We used default training parameters found in the repository and trained for 25 epochs. BLEU scores were calculated using the script from the repository and show similar convergence between BM8 and FP32 models.

---

[6]Implementation available at https://github.com/pytorch/fairseq

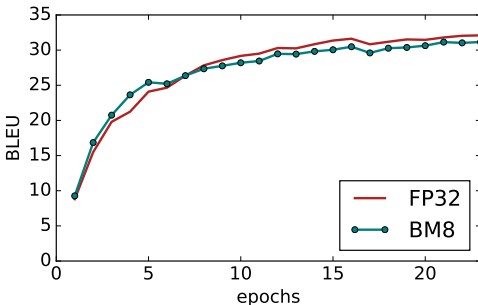

Figure 8: Training convergence curves for Transformer on IWSLT'14 DE-En dataset

### A.3.4   SSD-Lite (MobileNet-V2) (VOC)

We adapted a PyTorch implementation of SSD-lite from an online repository [7]. The base network is MobileNet-V2 (Sandler et al., 2018) which was pretrained on ImageNet. The enitre network is trained on VOC2012 and VOC2007 trainval datasets and evaluated on VOC2007 validation dataset. We apply BM8 quantization to all weights, activations and gradients before GEMM computations in the forward and backward paths. The network was trained with default parameter settings provided in the repository as follows: SGD with momentum of 0.9, weight decay factor 0.0005, batches of 32 images, and cosine annealing ($t_{max} = 200$) with an initial learning rate of 0.01. After 200 epochs, BM8 achieves a mAP of 68.0 which is sufficiently close to the reported accuracy of 68.6.

### A.3.5   EfficientNet-b6 (ImageNet)

We adapted a PyTorch implementation of EfficientNet (Tan & Le, 2019) from an online repository [8]. We trained the smallest EfficientNet-b0 network on a reduced sized ImageNet dataset, where the images are resized from 256x256 to 128x128. This choice was made to accelerate the training time, which is slowed down by $5\times$ with our BM8 quantization function. The network is trained on one GPU for only 60 epochs using batch size 256 and an initial learning rate of 0.1 which is decayed exponentially with gamma of 0.90387. Figure 9 shows convergence of BM8 with an FP32 baseline.

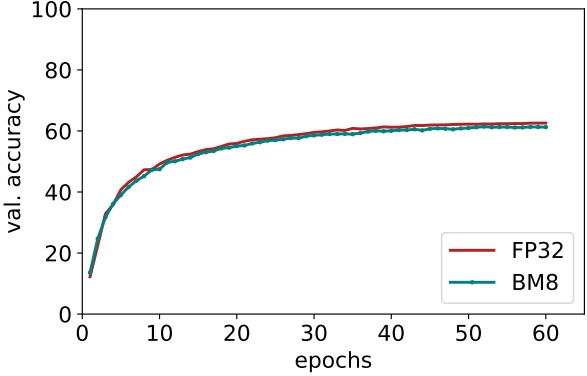

Figure 9: Training convergence curves for EfficientNet-b0 on ImageNet

### A.4   Hardware Synthesis

Fused multply-add (FMA) units were designed in RTL for floating-point and block minifloat representations. We modified code from the Deepfloat[9] repository for FP32, FP16 and FP8 units. The BM units with Kulisch accumulation were hand written in Verilog following the block design given

---

[7]Implementation available at https://github.com/qfgaohao/pytorch-ssd

[8]Implementation available at https://github.com/narumiruna/efficientnet-pytorch

[9]Implementation available at https://github.com/facebookresearch/deepfloat

in Figure 10. All designs were synthesised at 750Mhz using Cadence RTL compiler 14.11 and a commercial 28nm standard cell library. Since GEMM hardware is typically designed from tiles of smaller computational units, we also provide synthesis results for small 4x4 systolic array multipliers. Full coverage of our results are shown in Table 9 and Table 10. Table 11 is also provided, and shows the component breakdown and scaling of Kulisch related costs in different 8-bit BM regimes. Given that BM relies on hybrid formats, the multiplier operands are both sized for the largest mantissa plus one bit for denormal support. Compared to (4,3)/(5,2), which is the format used in HFP8 (Sun et al., 2019), BM8 (2,5)/(4,3) is $1.6\times$ smaller. This is because BM8 has narrower exponent encodings that reduces the width of the Kulisch accumulator. Finally, the overhead for converting from Kulisch to FP32 is relatively small. We synthesised this to only contribute $264\mu m^2$ in area for 32-bit accumulators, the cost of which is amortized over the length of the dot product.

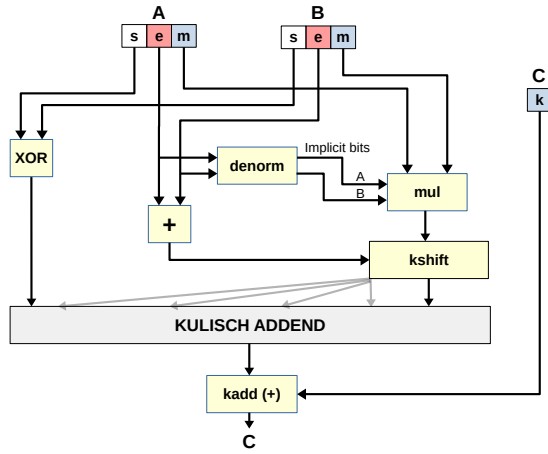

Figure 10: Block diagram of block minifloat multiply-add; A*B + C, where A and B are minifloats and C is an integer

Table 9: Synthesized logic area and power of single-cycle fused multiply-Add (FMA) at 750 MHz on 28nm chip.

| Component | Area $\mu m^2$ | Power $\mu W$ |
|---|---|---|
| FP32 | 4782 | 10051 |
| FP16 | 1116 | 2120 |
| FP8 (w/ FP16 add) | 829 | 1429 |
| INT8 (w/ INT32 add) | 417 | 1269 |
| BM8 | 391 | 1141 |
| BM7 | 280 | 840 |
| BM6 | 200 | 624 |
| BM5 | 171 | 546 |
| BM5-log | 231 | 801 |
| BM4 | 115 | 361 |
| BM4-log | 120 | 426 |

Table 10: Synthesized logic area and power of 4x4 systolic array multipliers at 750 MHz on 28nm chip.

| Component | Area $\mu m^2$ | Power $\mu W$ |
|---|---|---|
| FP8 (w/ FP16 add) | 18201 | 56202 |
| INT8 (w/ INT32 add) | 7005 | 20253 |
| BM8 | 6976 | 18765 |
| BM6 | 4083 | 11959 |

Table 11: Component breakdown and logic area for different 8-bit BM formats

| Format | Details | | | Area ($\mu m^2$) | | | |
|---|---|---|---|---|---|---|---|
| $(e, m)/(e, m)$ | multiply | kadd | kshift | comb.[1] | kadd | kshift | **total** |
| $(3, 4)/(4, 3)$ | $(5b \times 5b)$ | 34 | 24 | 210 | 93 | 74 | 377 |
| $(2, 5)/(4, 3)$ | $(6b \times 6b)$ | 31 | 20 | 235 | 79 | 77 | 391 |
| $(3, 4)/(5, 2)$ | $(5b \times 5b)$ | 49 | 40 | 253 | 199 | 95 | 547 |
| $(4, 3)/(5, 2)$ | $(4b \times 4b)$ | 56 | 48 | 259 | 276 | 104 | 639 |
| $(5, 2)/(5, 2)$ | $(3b \times 3b)$ | 71 | 64 | 300 | 361 | 113 | 774 |
| $(0, 7)/(0, 7)$ | $(8b \times 8b)$ | 32 | NA | NA | NA | NA | 418 |

[1] combinational logic includes multiply component

