# OpenReview forum: "A Block Minifloat Representation for Training Deep Neural Networks"
_ICLR.cc/2021/Conference — ICLR 2021 Poster_

### Official Review · AnonReviewer4 · 2020-10-27
**New numerical representations for training DNNs with very few bits, with impressive results. Some details need more explanation.**

**Rating:** 7
**Confidence:** 3

**Review:**

This paper proposes a family of numerical representations for training neural networks based on minifloats that share a common exponent across blocks. The authors perform a lot of software simulations to explore the design space on many different models and tasks. Hardware designs have also been synthesized and reported.

Pros:
-	The proposed representation is very general and covers a lot of existing designs, while also allowing for new ones.
-	An exhaustive exploration of the design space, and the discovery of some new low-precision representations that offer high accuracy as well as computational density.
-	A lot of different models and datasets are examined.

Cons
-	Some of the main contributions require a more careful, detailed explanation in order to fully appreciate and assess the contribution.
-	Results are presented in a confusing way.

Main areas for improvement:

One of the important contributions of the paper seems to be how the minifloat distribution is aligned with the value distribution and how $\beta$ is calculated to avoid underflow. However, this topic is only given a couple of sentences of explanation together with Figure 2, which does not provide much more information. I think the paper would be improved with a precise mathematical explanation of this re-alignment.

The block size for sharing exponents is determined semi-analytically by using Equation 6 to find a balance between the area cost and the dynamic range of the resulting numerical representation. However, this equation is introduced rather abruptly. The paper would be improved by providing more explanation of how Equation 6 was derived and some intuition into what the different terms mean. Furthermore, given the authors have implemented the block minifloat scheme in hardware, is it possible to show that Equation 6 actually matches what is seen when synthesizing the design?

The tables in Section 4, and the corresponding text, need some work. In Table 2, the footnote (2) is defined but seems to be unused. In Table 2, the different columns (w, x, dw, dx, acc) are not defined anywhere and it is also not clear what the footnote (2) means in this context since it applies to the ‘acc’ column across all schemes. Additionally, in the corresponding text the authors discuss the performance of BM7, but this scheme is not found in the table.


Additional comments and questions:

-	In equation (1) the sign bit $s$ is defined before the equation, but exponent $E$ and mantissa $F$ are not defined until the sentence afterwards.
-	In equation (2) it is not immediately clear what the index $i$ is meant to indicate. Furthermore, $X_i^a$ seems to indicate that the definition of $a$ depends on itself.
-	Aside from using minifloats, how does this work compare to Flexpoint [1], which takes a similar approach to shared exponents?
-	In Figure 6 – are the accuracy measurements actually taken from the hardware simulation?

[1] Köster, Urs, et al. "Flexpoint: An adaptive numerical format for efficient training of deep neural networks." Advances in neural information processing systems. 2017.

---

> ### Author Response · Authors · 2020-11-19
> **Response to "Official Review 3"**
>
> We would like to thank the reviewer for their valuable feedback and comments. We have uploaded a revised draft with more information and improved clarity based on suggestions and feedback received. Below, we provide answers and responses to specific questions and issues raised.
>
> #1. “alignment of minifloat distribution and value distribution requires more precise explanation”. This is indeed an important aspect of our work. In block minifloat, the shared exponent bias is calculated to align the minifloat distribution with the maximim exponent in the value distribution. The idea is to avoid overflow, which contributes the largest errors in dot-product computations. We have added Equation 3 in the revised draft to formalize this key step.
>
> #2. “Equation 6 requires more explanation”. This issue was also raised by Reviewer 2 and has been addressed in the revised draft, by Equation 7 now. The question remains how Equation 7 can be shown to match an implemented design. This is possible by defining the resource costs for each overhead (given by $\alpha$ in the text) using actual synthesized area results. For example, $\alpha_{1}$ would be equal to the area for BM multiply-add, and $\alpha_{2}$ would be the area of the floating-point hardware (which also includes the BM quantization and conversion modules). Since only $\alpha_{1}$ has been synthesized we can not use Equation 7 in a useful sense. Furthermore, Equation 7 is not designed to provide very accurate resource/area estimates since RTL synthesis tools are typically difficult to model. Rather, Equation 7 is expected to guide the choice of N, by capturing the main trend.
>
> #3. “Experiments and results are confusing”. This has been updated in the revised draft with appropriate definitions given for columns of the numerical representation (i.e. w, x, dw, dx). Additionally, BM7 has been added to the main results table from the Appendix.
>
> #4. “Notation issue with block minifloat definitions”. The original notation was indeed wrong. We have redefined $X_{i}^{a}$ to hopefully clear this up.
>
> #5. Flexpoint refers to a 16-bit block floating-point format with a 5-bit shared exponent. Block minifloats are considerably smaller, between 4 and 8 bits with an 8 bit shared exponent, and use minifloat rather than fixed-point as the underlying block representation. In terms of how the shared exponent is updated, Flexpoint proposes an algorithm called Autoflex to predict the output exponent, from a history of maximum exponents, and use the prediction to preemptively increase the exponent to avoid overflows. Our exponent update scheme is simpler and is calculated by the maximum exponent at that current point in training.
>
> #6. We simulate the behaviour of BM hardware using GPUs and Pytorch. Therefore, the accuracy is taken from ImageNet experiments running on the GPU. Our RTL code is only capable of simulating the BM dot-products, not the entire system.

---

### Official Review · AnonReviewer2 · 2020-10-28
**Simple extension of block floating point, interesting contribution on the hardware aspects**

**Rating:** 7
**Confidence:** 5

**Review:**

The authors proposes block-minifloat (BM), a floating-point format for DNN training. BM is a fairly simple extension to block floating-point (BFP), which was proposed in (Drumond 2018 and Yang 2019). In BFP, a block of integer mantissas share a single exponent. In BM, a block of narrow floats share a single exponent bias. The shared exponent bias helps to shorten the exponent field on each individual float element. This is a good contribution, though a bit trivial.

Where the paper make a strong contribution is in the hardware implementation of BM, something which neither Drumond or Yang really got into. The authors propose to use a Kulisch accumulator for minifloat dot products, which basically works by converting the floats to integers with a shift, and accumulating with a very wide register. Kulisch accumulators are normally far too wide to be practical (see Eq. 4), but they've been proposed for posit computation (https://engineering.fb.com/2018/11/08/ai-research/floating-point-math/). This seem like a great idea here since BM can reduce the exponent length to only 2 or 3 bits.

The authors also did a good job evaluating the area and power of the BM hardware circuit. They built a 4x4 systolic array multiplier using BM units in RTL, and synthesized to place and route. The results show that BM6 can be 4x smaller and use 2.3x less power than FP8 while achieving to comparable accuracy. This is a pretty impressive result, and the hardware evaluation methodology is more stringent than most quantization papers at NeurIPS/ICML/ICLR. The only **minor** issue I have here is the area/power numbers are reported for BM8/BM6, but the exact config is not specified. E.g. is BM8 referring to (2e, 5m)?

The accuracy comparison is pretty standard, with CIFAR and ImageNet results using mostly ResNet-18. The authors' simulation framework slows training by 5x, so this is as much as I would expect. One **major** issue is that Tables 1 and 3 shows that for training to succeed, the forward and backwards BM formats must be different. Table 3 has three separate BM formats for each row. Implementing them all in hardware could incur significant overhead, which the paper doesn't discuss. The authors mention that the HFP8 paper does the same - but that paper defends this practice by showing that their two formats (which only differ by 1 e/m bit) can be supported by a single FP unit with minimal overhead. This paper uses (2e,5m), (4e,3m) and (6e,9m) in the same experiment labeled "BM8", which seems both misleading and unjustified. Note that SWALP and S2FP8 (and bfloat16/float16 papers) would use the same format in forwards and backwards pass and avoid this overhead.

A few other insights: (1) subnormal floats are important and can't just be flushed to zero; (2) a square BM block size of 48x48 seems to work fine.

Minor issues:
 - The methodology for hardware area seem solid (Appendix 4), but there isn't much detail on power. Was power obtained through modeling or using an RTL simulator? What kind of test vectors were used?
 - The area/power numbers are given for "BM8", but what's the precise format? I assumed it was (2, 5).
 - The introduction of log-BM seems very sudden, and they're only used for VGG-16? Did regular BM5 not work? I'm not sure what to take away from the comparison in Table 2.
 - Equation 6 was a bit confusing for me. It would be helpful to explain briefly how each term was derived.
 - Training in BM requires you to update the exponent biases in each step (?), which requires computing the dynamic range of each $N \times N$ block. I believe this is probably negligible, but it should be discussed as an additional overhead.

EDIT: the authors have clarified that the hardware area results take into account the need to support multiple formats, which addressed my biggest issue with the paper. I have raised my score to a 7 (accept).

---

> ### Author Response · Authors · 2020-11-19
> **Response to "Official Review 2"**
>
> We would like to thank the reviewer for their valuable feedback and comments. We have uploaded a revised draft with more information and improved clarity based on suggestions and feedback received. Below, we provide answers and responses to specific questions and issues raised.
>
> 1. “minor issue about the specification of the BM configuration”. All BM formats are hybrid, meaning two formats are specified. BM8 always refers to (2,5)/(4,3), like BM6 always refers to (2,3)/(3,2). We’ve tried to clear this up in the text in “Section 3, Training with Block Minifloat, Hybrid representation”.
>
> 2. “major issue about three BM formats and associated overhead of the arithmetic unit”. Firstly, the (6e, 9) format is only applied to the weight gradients which does not require matrix multiplication. This means that the (6e, 9) does not involve computations mapped to block minifloat hardware (which is focused on dot-products in the GEMM). The issue now relates to the overhead for supporting (2e,5) and (4e,3) formats together, which have 2-bits mismatched. This primarily affects the mantissa multiplier, which must be large enough for the two largest mantissa operands. Put another way, the mantissa multiplier is not 5x3, rather it must support 5x5 (i.e. for weight*activation in forward path). Importantly, this has already been factored into our hardware as well as synthesis results. For some more clarity, Table 9 (original draft) or Table 11 (revised draft) provides component breakdowns of the logic area for (2e,5)/(4e,3) format and specify a 6x6 mantissa multiply. 6x6 instead of 5x5 because of the implicit bit which is added for supporting denormal numbers. We hope our explanation has cleared up this issue.
>
> 3.  “issue about power modelling”. The power comes from the Cadence RTL Compiler. It's important to note that these numbers are only rough estimates.
>
> 4. log-BM was included because we wanted to show that block minifloats cover the spectrum of formats between fixed-point and log representations at the corner cases. We compared VGG-16 with (Miyashita et al. 2016) since this is our only known example of logarithmic training. We haven’t tested BM5 or BM4 on VGG16, though we do expect similar if not better results than log-BM5 and log-BM4. This could be an interesting comparison point if we do in fact achieve better results (all be it on an outdated network). We have not had an opportunity to include this in our revised draft, but will consider the merits of this for any subsequent revision.
>
> 5. “issue about confusing Equation 6”. We agree. Our revised draft provides a much more thorough explanation for each term in what has now become Equation 7.

---

### Official Review · AnonReviewer1 · 2020-10-28
**A good submission but need to provide more information**

**Rating:** 6
**Confidence:** 4

**Review:**

This paper introduced a new representation (Block Minifloat) for training DNNs with low precisions of 8-bit or less. This new representation combines FP8 formats and the shared exponent bias concept to cover the dynamic range of tensors needed for DNN training. Compared to other published FP8 format, this representation has smaller exponents, which allows to use a more efficient Kulisch accumulator. The representation has been verified on a spectrum of deep learning models and datasets.

Overall, the paper is well written. The idea is clearly presented, and experiments are sound. Particularly, the hardware evaluation gives some impressive results. However, for the sake of clarity, I have some questions.

1). The shared-exponent bias itself is not new and use it for FP8 training is also straightforward and has limited novelty. What interesting is that the author uses this method to push for smaller exponent bits which in turn allows a more efficient accumulator. However, as a key contribution of this work, the authors did not give enough information and details on why “exponents up to 4 bits offer distinct advantages” for Kulishch accumulation. Could the authors explain more on the accumulator and provide some evident why smaller exponent is critical?

2). From emulation point of view, the authors use existing CUDA libraries for GEMM which basically uses standard FP32 floating point accumulation. When the authors use certain bit-width for Kulish accumulator elements (for e.g. Table 9 and Table 10), how do you know this accumulator setting won’t impact model convergence?

3). Hardware overhead for denorm support: the exponent bias is only guard against overflow, so denorm numbers are used to cover the range of smaller numbers. The authors claim that the hardware overhead is minimum since only input multiplicands are needed for the detection of denorm. However, there should be additional complexity than that. For example, how to handle denorm numbers in addend and how to tell the result produced is norm or denorm? Could the authors describe the hardware that needed to convert numbers from an intermediate format or a floating point format to their BM format?

4). For experiment, it would have been good to include some larger networks, such as ResNet50 on ImageNet to compared to SOTA.

---

> ### Author Response · Authors · 2020-11-19
> **Response to "Official Review 1"**
>
> We would like to thank the reviewer for their valuable feedback and comments. We have uploaded a revised draft with more information and improved clarity based on suggestions and feedback received. Below, we provide answers and responses to specific questions and issues raised.
>
> #1. The main reason for  “up to 4-bit exponents” is because in these regimes block minifloat representations (with Kulisch accumulation) have INT8-like hardware complexity  (i.e. 8-bit multiply and 32-bit accumulate). INT8 is important because it represents the most efficient arithmetic scheme capable of training tasks like ImageNet to an accuracy that is close to FP32. Put another way, the Kulisch accumulator is prohibitively wide and expensive for formats with larger exponents. This becomes clear when comparing kadd for different formats, but more importantly, it is supported by our hardware synthesis results of different formats. We have updated “Section 2.3 Kulisch Accumulation” in the revised draft with a similar explanation. Also, we have included Table 11 in the Appendix which provides a better comparison for kadd and related area costs.
>
> #2. The Kulisch accumulator is always sized to calculate an error free sum of products. On the otherhand, FP32 accumulators may produce small rounding errors when adding very large and small numbers together. In such cases, the accumulator may not have enough mantissa bits (i.e. 23 in FP32) to tolerate large right shifts. Given that most DNNs train successfully with FP32 accumulation, it is reasonable to assume that FP32 is a very close approximation to an error-free Kulisch accumulator.
>
> #3. "issue regarding denormal overheads". In our estimation, the hardware overhead for supporting denormal numbers is small. Only one extra bit (i.e. the implicit bit) must be incorporated into each mantissa for multiplication. This does incur some overhead, though the resultant arithmetic units are typically still very small. We cite our synthesis results to support this claim. Furthermore, there is no additional complexity in the addend or result of the MAC unit because these are represented in fixed point. To determine whether a partial sum (for a given dot product) is norm or denorm, the value must be converted to a floating point representation, using standard fixed to float conversion hardware, where the number will be denorm if the floating-point exponent encoding is 0 (as in Equation 1).
>
> #3. Conversion from fixed point to floating-point (and subsequently Block Minifloat) involves the following steps. 1.) A leading-one detection module is required to determine the position of the most significant bit, 2.) the fixed point number must be shifted so the decimal point is at the most significant bit, 3.) the position (which is a bit index and location of decimal point) should be compared to the representable range of the floating point or minifloat format. This involves conditional logic to determine whether the number should be saturated, flushed-to-zero, denorm or norm. 4.) A denorm number exists if the position is equal to the minimum exponent in the minifloat format (after the bias applied). In such cases, the decimal point must be left shifted by one more bit, 5.) The last step involves quantization of the fractional bits (or mantissa) using stochastic rounding. This can be implemented with a multiplier and linear feedback shift register (LFSR). We have provided a little bit of context to this description in “Section 5 Hardware Evaluation” of the revised draft. This is to supplement the main analysis of block minifloat arithmetic units.
>
> 4.) This was the plan. We started ResNet50, but the training time proved to be a bottleneck for effective testing and exploration. We instead focused on smaller networks that we feel are more suitable to the targeted embedded domain anyway, where stricter computational power and area constraints exist.

---

### Decision · Program_Chairs · 2021-01-07
**Final Decision**

**Decision:**

Accept (Poster)

**Comment:**

This paper proposes a new approach to training networks with low precision called Block Minifloat. The reviewers found the paper well written and found that the empirical results were sufficient. In particular, they found the hardware implementation was a strong contribution. Furthermore, the rebuttal properly addressed the comments of the reviewer.